# General framework for E(3)-equivariant neural network representation of density functional theory Hamiltonian

Xiaoxun Gong [1,2,7], He Li [1,3,4,7], Nianlong Zou[1], Runzhang Xu[1], Wenhui Duan [1,3,4,5] ✉ & Yong Xu [1,4,5,6] ✉

The combination of deep learning and ab initio calculation has shown great promise in revolutionizing future scientific research, but how to design neural network models incorporating a priori knowledge and symmetry requirements is a key challenging subject. Here we propose an E(3)-equivariant deep-learning framework to represent density functional theory (DFT) Hamiltonian as a function of material structure, which can naturally preserve the Euclidean symmetry even in the presence of spin–orbit coupling. Our DeepH-E3 method enables efficient electronic structure calculation at ab initio accuracy by learning from DFT data of small-sized structures, making the routine study of large-scale supercells (>$10^4$ atoms) feasible. The method can reach sub-meV prediction accuracy at high training efficiency, showing state-of-the-art performance in our experiments. The work is not only of general significance to deep-learning method development but also creates opportunities for materials research, such as building a Moiré-twisted material database.

It has been well recognized that deep-learning methods could offer a potential solution to the accuracy-efficiency dilemma of ab initio material calculations. Deep-learning potential[1,2] and a series of other neural network models[3–7] are capable of predicting the total energies and atomic forces of given material structures, enabling molecular dynamics simulation at large length and time scales. The paradigm has been used for deep-learning research of various kinds of physical and chemical properties[8–19]. During the development of these methods, people have gradually come to realize that the introduction of symmetry considerations as a priori knowledge into neural networks is of crucial importance to the deep-learning approaches. For this purpose, people have drawn insights from a class of neural networks called the equivariant neural networks (ENNs)[20–24]. The key innovation of ENNs is that all the internal features transform under the same symmetry group with the input; thus, the symmetry requirements are explicitly treated and exactly satisfied. Symmetry has fundamental importance in physics, so ENNs will be especially advantageous when they are applied to the modeling of physical systems, as shown by a series of neural network models for various material properties[6,7,13–15].

Recently, a deep neural network representation of density functional theory (DFT) Hamiltonian (named DeepH) was developed by employing the locality of electronic matter, localized basis, and local coordinate transformation[25]. By the DeepH approach, the computationally demanding self-consistent field iterations could be bypassed, and all the electron-related physical quantities in the single-particle picture can, in principle, be efficiently derived. This opens opportunities for the electronic structure calculation of large-scale material systems. However, it is highly nontrivial to incorporate symmetry considerations into DeepH. Specifically, the property that the Hamiltonian matrix changes covariantly (i.e., equivariantly) under rotations or gauge transformations should be preserved by the neural network model for efficient learning and accurate prediction (Fig. 1). A strategy is developed in DeepH to apply local coordinate transformation which changes the rotation covariant problem into an invariant one and thus

[1]State Key Laboratory of Low Dimensional Quantum Physics and Department of Physics, Tsinghua University, 100084 Beijing, China. [2]School of Physics, Peking University, 100871 Beijing, China. [3]Institute for Advanced Study, Tsinghua University, 100084 Beijing, China. [4]Tencent Quantum Laboratory, Tencent, 518057 Shenzhen, Guangdong, China. [5]Frontier Science Center for Quantum Information, Beijing, China. [6]RIKEN Center for Emergent Matter Science (CEMS), Wako, Saitama 351-0198, Japan. [7]These authors contributed equally: Xiaoxun Gong, He Li. ✉e-mail: duanw@tsinghua.edu.cn; yongxu@mail.tsinghua.edu.cn

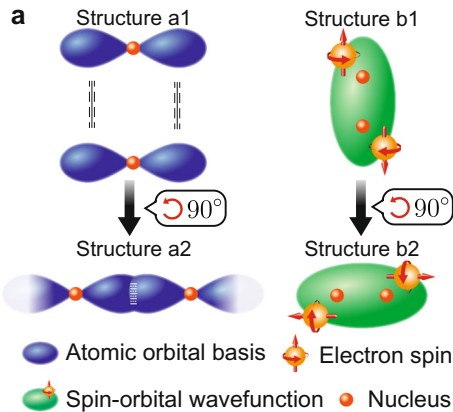

**Fig. 1 | Equivariance in electronic structure calculations. a** Schematic wave-functions and **b** Hamiltonian matrices are shown for the systems neglecting or including spin–orbit coupling (SOC). Structures a1 and a2 are related to each other by a 90° rotation (see (**a**)), whose hopping parameters (i.e., Hamiltonian matrix elements) between $p_x$ orbitals are related by a unitary transformation (see (**b**)). This equivariant property of Hamiltonian must be preserved in all electronic structure calculations. When the SOC is taken into account, the spin and orbital degrees of freedom are coupled and must transform together under global rotations, as shown for structures b1 and b2.

the transformed Hamiltonian matrices can be learned flexibly via rotation-invariant neural networks[25]. Nevertheless, the large amount of local coordinate information seriously increases the computational load, and the model performance depends critically on a proper selection of local coordinates, which relies on human intuition and is not easy to optimize. Therefore, we think that the combination of DeepH with ENN might open new possibilities for the deep-learning modeling of Hamiltonians.

People have already made attempts to model the Hamiltonian using equivariant methods. Unke et al. designed PhiSNet[26], which used ENN for predicting the Hamiltonians of molecules with fixed system size. Nigam et al. used rotationally equivariant N-center features in the kernel ridge regression method to fit molecular Hamiltonians[27]. Zhang et al. proposed an equivariant scheme to parameterize the Hamiltonians of crystals based on the atomic cluster expansion descriptor[28]. However, the key capability of DeepH that learns from DFT results on small-sized material systems and predicts the electronic structures of much larger ones has not been demonstrated by these methods. More critically, the existing equivariant methods have neglected the equivariance in the spin degrees of freedom, although the electronic spin and spin–orbit coupling (SOC) play a key role in modern condensed matter physics and materials science. With SOC, one should take care of the spin–orbital Hamiltonian, whose spin and orbital degrees of freedom are coupled and transform together under a change of coordinate system or basis set, as illustrated in Fig. 1. This would raise critical difficulties in designing ENN models due to a fundamental change of symmetry group. In this context, the incorporation of ENN models into DeepH is essential but remains elusive.

In this work, we propose DeepH-E3, a universal E(3)-equivariant deep-learning framework to represent the spin–orbital DFT Hamiltonian $\hat{H}_{DFT}$ as a function of atomic structure $\{\mathcal{R}\}$ by neural networks, which enables efficient electronic structure calculations of large-scale materials at ab initio accuracy. A general theoretical basis is developed to explicitly incorporate covariance transformation requirements of $\{\mathcal{R}\} \mapsto \hat{H}_{DFT}$ into neural network models that can properly take the electronic spin and SOC into account, and a code implementation of DeepH-E3 based on the message-passing neural network is also presented. Since the principle of covariance is automatically satisfied, efficient learning and accurate prediction become feasible via the DeepH-E3 method. Our systematic experiments demonstrate the state-of-the-art performance of DeepH-E3, which shows sub-meV accuracy in predicting DFT Hamiltonian. The method works well for various kinds of material systems, such as magic-angle twisted bilayer graphene or twisted van der Waals materials in general, and the

computational costs are reduced by several orders of magnitude compared to direct DFT calculations. Benefiting from the high efficiency and accuracy as well as the good transferability, there could be promising applications of DeepH-E3 in electronic structure calculations. Also, we expect that the proposed neural network framework can be generally applied to develop deep-learning ab initio methods and that the interdisciplinary developments would eventually revolutionize future materials research.

## Results
### Realization of equivariance
It has long been established as one of the fundamental principles of physics that all physical quantities must transform equivariantly between reference frames. Formally, a mapping $f : X \to Y$ is equivariant for vector spaces $X$ and $Y$ with respect to group $G$ if $D_Y(g) \circ f = f \circ D_X(g), \forall g \in G$, where $D_X, D_Y$ are representations of group $G$ over vector spaces $X, Y$, respectively. The problem considered in this work is the equivariance of a mapping from the material structure $\{\mathcal{R}\}$ including atom types and positions to the DFT Hamiltonian $\hat{H}_{DFT}$ with respect to the E(3) group. The E(3) group is the Euclidean group in three-dimensional (3D) space which contains translations, rotations, and inversion. Translation symmetry is manifest since we only work on the relative positions between atoms, not their absolute positions. Rotations of coordinates introduce nontrivial transformations, which should be carefully investigated. Suppose the same point in space is specified in two coordinate systems by **r** and **r**′. If the coordinate systems are related to each other by a rotation, the transformation rule between the coordinates of the point is **r**′ = **Rr**, where **R** is a 3 × 3 orthogonal matrix.

In order to take advantage of the nearsightedness of electronic matter[29], the Hamiltonian operator is expressed in the picture of localized pseudo-atomic orbital (PAO) basis. The basis is separated into radial and angular parts, having the form $\phi_{i\alpha}(\mathbf{r}) = R_{ipl}(r)Y_{lm}(\hat{r})$. Here $i$ is the site index, $\alpha \equiv (plm)$, where $p$ is the multiplicity index, $Y_{lm}$ is the spherical harmonics having angular momentum quantum number $l$ and magnetic quantum number $m$, $r \equiv |\mathbf{r} - \mathbf{r}_i|$ and $\hat{r} \equiv (\mathbf{r} - \mathbf{r}_i)/|\mathbf{r} - \mathbf{r}_i|$ where $\mathbf{r}_i$ is the position of the $i$th atom. The transformation rule for the Hamiltonian matrix between the two coordinate systems described above is

$$\left(H'_{ip_1jp_2}\right)^{l_1 l_2}_{m_1 m_2} = \sum_{m'_1 = -l_1}^{l_1} \sum_{m'_2 = -l_2}^{l_2} D^{l_1}_{m_1 m'_1}(\mathbf{R}) D^{l_2}_{m_2 m'_2}(\mathbf{R})^* \left(H_{ip_1jp_2}\right)^{l_1 l_2}_{m'_1 m'_2}, \quad (1)$$

where $D^l_{mm'}(\mathbf{R})$ is the Wigner D-matrix. The equivariance of the mapping $\{\mathcal{R}\} \mapsto \hat{H}_{\text{DFT}}$ requires that, if the change of coordinates causes the positions of the atoms to transform, the corresponding Hamiltonian matrix must transform covariantly according to Eq. (1).

ENN is applied to construct the mapping $\{\mathcal{R}\} \mapsto \hat{H}_{\text{DFT}}$ in order to preserve equivariance. The input, output, and internal feature vectors of ENNs all belong to a special set of vectors that have the form $\mathbf{x}_l = (x_{l,l}, \ldots, x_{l,-l})$ and transform according to the following rule:

$$x'_{lm} = \sum_{m'=-l}^{l} D^l_{mm'}(\mathbf{R})x_{lm'}. \tag{2}$$

This vector is said to carry the irreducible representation of the SO(3) group of dimension $2l+1$. If the input vectors are transformed according to Eq. (2), then all the internal features and the output vectors of the ENN will also be transformed accordingly. Under this constraint, the ENN incorporates learnable parameters in order to model equivariant relationships between inputs and outputs.

The method of constructing the equivariant mapping $\{\mathcal{R}\} \mapsto \hat{H}_{\text{DFT}}$ is illustrated in Fig. 2. The atomic numbers $Z_i$ and interatomic distances $|\mathbf{r}_{ij}| \equiv |\mathbf{r}_i - \mathbf{r}_j|$ are used to construct the $l=0$ input vectors (scalars). Spherical harmonics acting on the unit vectors of relative positions $\hat{r}_{ij}$ constitute input vectors of $l = 1, 2, \ldots$. The output vectors of the ENN are passed through the Wigner–Eckart layer before representing the final Hamiltonian. This layer exploits the essential concept of the Wigner–Eckart theorem:

$$l_1 \otimes l_2 = |l_1 - l_2| \oplus \cdots \oplus (l_1 + l_2). \tag{3}$$

"$\oplus$" and "$\otimes$" signs stand for direct sum and tensor product of representations, respectively. "=" denotes equivalence of representations, i.e., they differ from each other by a change of basis. The coefficients in the change of basis are exactly the celebrated Clebsch–Gordan coefficients. The representation $l_1 \otimes l_2$ is carried by

the tensor $\mathbf{x}_{l_1 l_2}$, which transforms according to the rule

$$x'_{l_1 l_2 m_1 m_2} = \sum_{m'_1 = -l_1}^{l_1} \sum_{m'_2 = -l_2}^{l_2} D^{l_1}_{m_1 m'_1}(\mathbf{R}) D^{l_2}_{m_2 m'_2}(\mathbf{R}) x_{l_1 l_2 m'_1 m'_2}. \tag{4}$$

Notice that Eq. (4) has the same form as Eq. (1), so the tensor $\mathbf{x}_{l_1 l_2}$ can exactly represent the output Hamiltonian satisfying the equivariant requirements.

### Equivariance of the spin–orbital Hamiltonian

If we further consider the spin degrees of freedom, the transformation rule for the Hamiltonian becomes

$$\left( H'_{ip_1 jp_2} \right)^{l_1 \frac{1}{2} l_2 \frac{1}{2}}_{m_1 \sigma_1 m_2 \sigma_2} =$$
$$\sum_{m'_1 = -l_1}^{l_1} \sum_{m'_2 = -l_2}^{l_2} \sum_{\sigma'_1 = \uparrow, \downarrow} \sum_{\sigma'_2 = \uparrow, \downarrow} D^{l_1}_{m_1 m'_1}(\mathbf{R}) D^{\frac{1}{2}}_{\sigma_1 \sigma'_1}(\mathbf{R})$$
$$D^{l_2}_{m_2 m'_2}(\mathbf{R})^* D^{\frac{1}{2}}_{\sigma_2 \sigma'_2}(\mathbf{R})^* \left( H_{ip_1 jp_2} \right)^{l_1 \frac{1}{2} l_2 \frac{1}{2}}_{m'_1 \sigma'_1 m'_2 \sigma'_2}, \tag{5}$$

where $\sigma_1, \sigma_2$ are the spin indices (spin up or down). The construction of the spin–orbital DFT Hamiltonian is a far more complicated issue. Electron spin has angular momentum $l = 1/2$, so it seems that tedious coding and debugging are unavoidable because we have to introduce complex-valued half-integer representations into the neural network, which typically only supports real-valued integer representations for the time being. Furthermore, a $2\pi$ rotation brings a vector in 3D space to itself but introduces a factor -1 to the spin-1/2 vector. This means that any mapping from 3D input vectors to $l = 1/2$ output vectors will be discontinued and cannot be modeled by neural networks, which poses a serious threat to our approach since we only have 3D vectors as input to the neural network (Fig. 2).

Fortunately, we observe that $l = 1/2$ appearing in the DFT Hamiltonian does not necessarily mean that half-integer representations

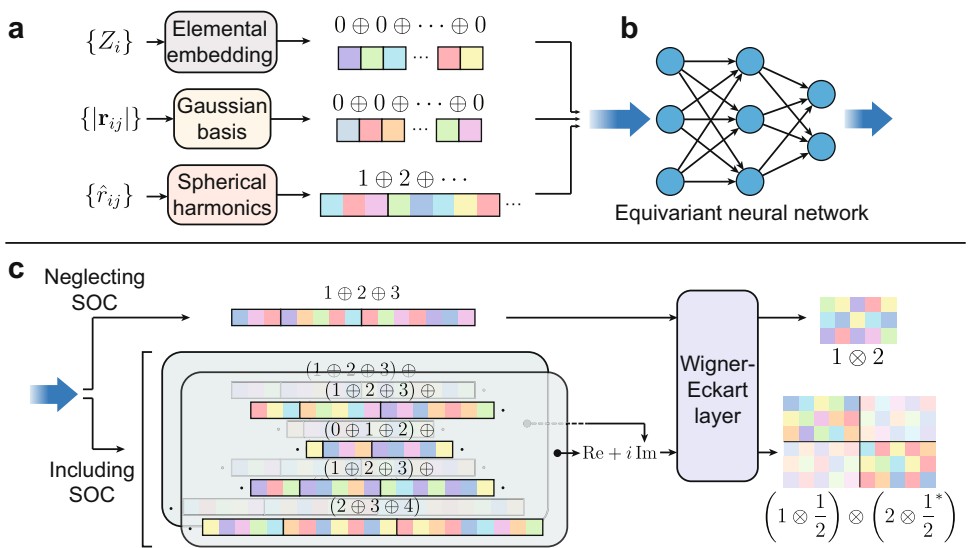

**Fig. 2 | Method of constructing an equivariant mapping $\{\mathcal{R}\} \mapsto \hat{H}_{\text{DFT}}$.** Take the Hamiltonian matrix between $l = 1$ and $l = 2$ orbitals, for example. **a** The atomic numbers $Z_i$ and interatomic distances $|\mathbf{r}_{ij}|$ are used to construct the $l = 0$ vectors, and the unit vectors of relative positions $\hat{r}_{ij}$ are used to construct vectors of $l = 1, 2, \ldots$. **b** These vectors are passed to the equivariant neural network. **c** If neglecting spin–orbit coupling (SOC), the output vectors of the neural network are converted to the Hamiltonian using the rule $1 \oplus 2 \oplus 3 = 1 \otimes 2$ via the Wigner–Eckart layer. If including SOC, the output consists of two sets of real vectors which are combined to form complex-valued vectors. These vectors are converted to the spin–orbital DFT Hamiltonian according to a different rule $(1 \oplus 2 \oplus 3) \oplus (0 \oplus 1 \oplus 2) \oplus (1 \oplus 2 \oplus 3) \oplus (2 \oplus 3 \oplus 4) = (1 \otimes \frac{1}{2}) \otimes (2 \otimes \frac{1}{2}^*)$.

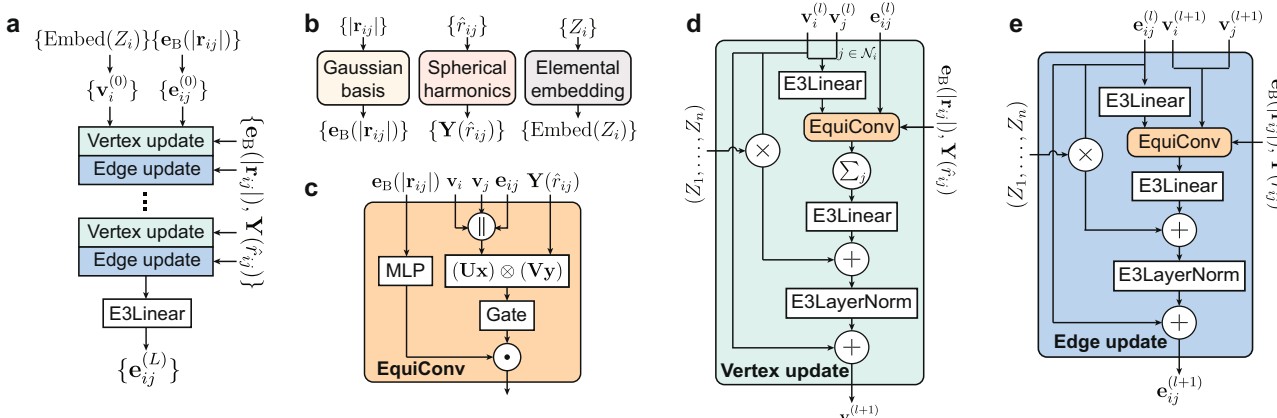

**Fig. 3 | Neural network architecture of DeepH-E3. a** Overall network structure. Elemental embeddings and Gaussian expansions (see (**b**)) serve as initial vertex and edge features, respectively. The vertex and edge features are updated $L$ times by update blocks (see (**d**) and (**e**)), which encode the interatomic distances and directional information through equivariant convolutions (EquiConv, see (**c**)). The "·" sign stands for channel-wise multiplication and || for vector concatenation. $\mathcal{N}_i$ is the neighborhood of vertex $i$. The final edge vectors $\{\mathbf{e}_{ij}^{(L)}\}$ are passed into the Wigner–Eckart layer depicted in Fig. 2 to represent the DFT Hamiltonian.

must be inserted everywhere into the neural network. In fact, they can be restricted to the final output layer as soon as we employ the transformation rule:

$$\left(l_1 \otimes \frac{1}{2}\right) \otimes \left(l_2 \otimes \frac{1}{2}\right) = (l_1 \otimes l_2) \otimes (0 \oplus 1). \tag{6}$$

There is no half-integer representation on the right-hand side; thus, it can be further decomposed into integer representations by repeatedly applying Eq. (3).

Another problem is associated with the introduction of complex numbers. Generally, the spin–orbital Hamiltonian matrix elements have complex values, and the ENN cannot simply predict its real and imaginary parts separately because this will violate equivariance. Ordinary neural networks of complex numbers are mostly still under their experimental and developmental stage, so the use of complex-valued ENN is practically difficult, if not impossible. Nevertheless, we have discovered a way to sidestep this problem. Under the bases which are eigenvectors of the time-reversal operator, the D-matrices of integer $l$ will become purely real. Consequently, for a vector with integer $l$ under that basis, its complex and real parts will never mingle with each other when the vector is multiplied by a real transformation matrix. Then one complex vector can be technically treated as two real vectors while preserving equivariance. Note that this is not true for half-integer representations, for that we must add up the real and imaginary parts before the integer representations are converted to half-integer representations in the Wigner–Eckart layer (Fig. 2).

Yet another subtle issue arises in Eq. (5). It is not exactly the same as Eq. (4) in that two of the D-matrices in the former equation are taken as complex conjugates, but those in the latter are not. In fact, instead of constructing a vector with representation $(l_1 \otimes \frac{1}{2}) \otimes (l_2 \otimes \frac{1}{2})$, we must construct $(l_1 \otimes \frac{1}{2}) \otimes (l_2^* \otimes \frac{1}{2}^*)$ to represent the spin–orbital Hamiltonian described in Eq. (5). Here, $l^*$ denotes the representation whose representation matrix is replaced by its complex conjugate. This is not a problem for integer $l$, but is critical for $l = 1/2$. If not treated properly, the overall equivariance will be violated. In order to solve this problem, we first notice that the representation $l^*$ is still a representation of the SU(2) group with dimension $2l + 1$. In fact, it is guaranteed to be equivalent to the representation $l$ without complex conjugate. In other words, there must exist a unitary matrix $\mathbf{P}^l$ for each integer or half-integer $l$ satisfying

$$\mathbf{D}^l(g)^* = \mathbf{P}^l \mathbf{D}^l(g)(\mathbf{P}^l)^\dagger, \forall g \in \text{SU}(2). \tag{7}$$

This is guaranteed by the fact that the quantum rotation operator $\hat{U}(g)$ commutes with the time-reversal operator $\mathcal{T}$: $\langle lm|\hat{U}(g)|lm'\rangle = \langle lm|\mathcal{T}^\dagger \hat{U}(g)\mathcal{T}|lm'\rangle = (-1)^{m-m'}\langle l, -m|\hat{U}(g)|l, -m'\rangle^*$. The matrix $\mathbf{P}$ in Eq. (7) is thus given by

$$P_{mm'}^l = (-1)^{l-m}\delta_{m,-m'}. \tag{8}$$

Therefore, we only need to apply a change of basis to convert a vector carrying representation $l$ to a vector carrying $l^*$. Notice that this property holds even for material systems without time-reversal symmetry.

The workflow of constructing the DFT Hamiltonian is summarized and illustrated in Fig. 2. In order to construct a Hamiltonian with SOC, the output vectors from the ENN are first separated into two real components, then combined together into complex vectors and passed to the Wigner–Eckart layer. The Wigner–Eckart layer uses the rules in Eq. (3) and Eq. (6) to convert these vectors to tensors of the form in Eq. (4), except that the tensors here have rank 4 instead of 2. After that, the last spin index is converted to its complex conjugate counterpart by the change of basis using Eq. (8) for $l = 1/2$. The output tensors follow the same transformation rule under coordinate rotation as the DFT Hamiltonian in Eq. (5), and thus could be used to represent the DFT Hamiltonian matrix.

Finally, we discuss two remaining issues. To include parity, we will consider $\text{E}(3) = \text{SE}(3) \otimes \{E, I\}$, where $E$ is the identity and $I$ is the spatial inversion. Under a coordinate transform, the vector is multiplied by $-1$ if it has odd parity and the coordinate transform involves spatial inversion. The parity of the Hamiltonian is determined by $(-1)^{l_1 + l_2}$. In addition, there is a possible ambiguity in Eq. (5) since the mapping from a classical rotation $\mathbf{R}$ to a quantum rotation $\mathbf{D}^{\frac{1}{2}}$ is not single-valued. However, the possible factor $-1$ will always be canceled between the two D-matrices in that equation, which eliminates the potential problem.

## The neural network architecture of DeepH-E3

Here we present the neural network architecture of the DeepH-E3 method. An illustration of the architecture can be found in Fig. 3. The general structure is based on the message-passing neural network[9,30] that has been widely used in materials research[6,7,14–19,25]. The material structure is represented by a graph, where each atom is associated with a vertex (or node). Edges are connected between atom pairs with nonzero inter-site hopping, and self-loop edges are included to describe intra-site coupling. Every vertex $i$ is associated with a feature $\mathbf{v}_i$ and every edge $ij$ with $\mathbf{e}_{ij}$. These features are composed of several

**Table 1 | MAEs of DFT Hamiltonian matrix elements averaged over atom pairs for monolayer graphene and MoS$_2$, all in units of meV.$^a$**

|  | Graphene | MoS$_2$ | | | |
|---|---|---|---|---|---|
|  | C-C | Mo-Mo | Mo-S | S-Mo | S-S |
| DeepH | 2.1 | 1.3 | 1.0 | 0.8 | 0.7 |
| DeepH-E3 | **0.40** | **0.51** | **0.46** | **0.45** | **0.37** |

$^a$The best result of each target is marked as bold. For MoS$_2$, there are four different types of atom pairs whose MAEs are listed separately.

vectors defined in Eq. (2). As illustrated in Fig. 3a, the initial feature $\mathbf{v}_i^{(0)}$ of vertex $i$ is the trainable embedding of the atomic number $Z_i$, and the initial $\mathbf{e}_{ij}$ is the interatomic distance $|\mathbf{r}_{ij}|$ expanded using the Gaussian basis $\mathbf{e}_B(|\mathbf{r}_{ij}|)$ as defined in Eq. (11). The features of vertices and edges are iteratively updated using features of their neighborhood as incoming messages. Finally, the final edge feature $\mathbf{e}_{ij}$ is passed through a linear layer and used to construct the Hamiltonian matrix block $H_{ij}$ between atoms $i$ and $j$ using the method illustrated in Fig. 2. It is worth mentioning that, under the message-passing scheme, the output Hamiltonian is only influenced by the information of its neighborhood environment. The nearsightedness property[29] ensures efficient linear-scaling calculations as well as good generalization ability[25].

The equivariant building blocks of the neural network are implemented using the scheme provided by Tensor-Field Networks[21] and e3nn[24,31]. The feature vectors $x_{cm}^{(l)}$ processed by these neural network blocks are implemented as dictionaries with key $l$, an integer which is the order of representation of the SO(3) group. $c$ is the "channel index" ranging from 1 to $n^{(l)}$, where $n^{(l)}$ is the number of channels at order $l$, and each channel refers to a vector defined in Eq. (2).

The E3Linear layer defined in Eq. (12) possesses learnable weights and biases, which is similar to linear layers in conventional neural networks, but only connects vectors of the same representation to preserve equivariance. The gate layer introduces equivariant non-linearity, as proposed in ref. 22, where nonlinearly activated $l = 0$ vectors (i.e., scalars) are used as scaling factors ("gates") to the norms of $l \neq 0$ vectors.

We propose a normalization scheme, E3LayerNorm, that normalizes the feature vectors using mean and variance obtained from the layer statistics while preserving equivariance:

$$\text{E3LayerNorm}\,(\mathbf{v}_i)_{cm}^{(l)} = g_c^{(l)} \frac{(\mathbf{v}_i)_{cm}^{(l)} - \mu_m^{(l)}}{\sigma^{(l)} + \epsilon} + b_c^{(l)}, \qquad (9)$$

where $\epsilon$ is introduced to maintain numerical stability, $g_c^{(l)}, b_c^{(l)}$ are learnable affine parameters, the mean $\mu_m^{(l)} = \frac{1}{Nn^{(l)}} \sum_{i=1}^{N} \sum_{c=1}^{n^{(l)}} (\mathbf{v}_i)_{cm}^{(l)}$, the variance $(\sigma^{(l)})^2 = \frac{1}{Nn^{(l)}} \sum_{i=1}^{N} \sum_{c=1}^{n^{(l)}} \sum_{m=-l}^{l} |(\mathbf{v}_i)_{cm}^{(l)} - \mu_m^{(l)}|^2$, $N$ is the total number of vertices. Here only the E3LayerNorm for vertex update blocks is described. The corresponding E3LayerNorm for edge update blocks is similar with the mean and variance obtained from edge features instead of vertex features. We find that E3LayerNorm significantly stabilizes the training process. A discussion about the use of E3LayerNorm can be found in Supplementary Note 5.

The previously discussed blocks do not include coupling between different $l$'s. This problem is resolved by the tensor product layer:

$$z_{cm}^{(l)} = \sum_{l_1 l_2} \sum_{m_1 m_2} \sum_{c_1 c_2} C_{l_1 m_1 ; l_2 m_2}^{lm} \left( U_{cc_1}^{(l_1)} x_{c_1 m_1}^{(l_1)} \right) \left( V_{cc_2}^{(l_2)} y_{c_2 m_2}^{(l_2)} \right), \qquad (10)$$

where $C_{l_2 m_2 ; l_3 m_3}^{l_1 m_1}$ are Clebsch–Gordan coefficients, $U_{cc'}^{(l)}, V_{cc'}^{(l)}$ are learnable weights. This is abbreviated as $\mathbf{z} = (\mathbf{U}x) \otimes (\mathbf{V}y)$.

The neural network architecture is illustrated in Fig. 3. The equivariant convolution block (EquiConv, Fig. 3c) encodes the information of an edge and the vertices connected to that edge. The core component of equivariant convolution is the tensor product (Eq. (10)) of the vertex and edge features ($\mathbf{v}_i \| \mathbf{v}_j \| \mathbf{e}_{ij}$) and the spherical harmonics of the edge $ij$ ($\mathbf{Y}(\hat{r}_{ij})$). Here $\|$ stands for vector concatenation. The tensor product introduces directional information of material structure into the neural network. Propagating directional information into neural networks is important, as emphasized by previous works[12,14], which is realized in an elegant way here via the tensor product. The interatomic distance information is also encoded into the neural network. It is expanded using the Gaussian basis expansion and then fed into a fully connected neural network, whose output is multiplied element-wise to the output of gate nonlinearity.

The vertex update block (Fig. 3d) aggregates information from the neighboring environment. To update a vertex, every edge connected to that vertex contributes a "message" generated by the equivariant convolution (EquiConv) block. All the "messages" are summed and normalized to update the vertex feature. This is similar for the edge update block (Fig. 3e), except that only the output of EquiConv on edge $ij$ is used for updating $\mathbf{e}_{ij}$. After several updates, the final edge feature vectors will serve as the neural network output and are passed into the Wigner–Eckart layer to construct the Hamiltonian matrix blocks, as illustrated in Fig. 2. More details are described in "Methods".

## Capability of DeepH-E3

The incorporation of global Euclidean symmetry as a priori knowledge provided to the message-passing deep-learning framework in the DeepH-E3 model has led to its outstanding performance in terms of efficiency and accuracy. A remarkable capability of DeepH-E3 is to learn from DFT data on small structures and make predictions on varying structures of different sizes without having to perform further DFT calculations. This enables highly efficient electronic structure calculations of large-scale material systems at ab initio accuracy. All the DFT Hamiltonian matrices used for deep learning in this work are computed by the OpenMX code using the PAO basis. After example studies on monolayer graphene and MoS$_2$ datasets, we will first demonstrate the capability of DeepH-E3 by investigating twisted bilayer graphene (TBG), especially the well-known magic-angle TBG whose DFT calculation is important but quite challenging due to its huge Moiré supercell. Next, we will apply DeepH-E3 to study twisted van der Waals (vdW) materials with strong SOC, including bilayers of bismuthene, Bi$_2$Se$_3$, and Bi$_2$Te$_3$, for demonstrating the effectiveness of our equivariant approach to construct the spin–orbital DFT Hamiltonian. Finally, we will use our model to illustrate the SOC-induced topological quantum phase transition in twisted bilayer Bi$_2$Te$_3$, giving an example of exploring exotic physical properties in large-scale material systems.

## Study of monolayer graphene and MoS$_2$

Before going to large-scale materials, we first validate our method on the datasets used in ref. 25 to benchmark DeepH-E3's performance. The datasets are comprised of DFT supercell calculation results of monolayer graphene and MoS$_2$, and different geometric configurations are sampled from ab initio molecular dynamics. The test results are summarized in Table 1 and compared with those of the original DeepH method[25], which, instead of using an explicitly equivariant approach, applied the local coordinate technique in handling the covariant transformation property of the Hamiltonian. Our experiments show that the mean absolute errors (MAEs) of Hamiltonian matrix elements averaged over atom pairs are all within a fraction of a meV, which are reduced approximately by a factor of 2 or more in all prediction targets compared with DeepH. Benefiting from the high accuracy of the deep-learning DFT Hamiltonian, band structures

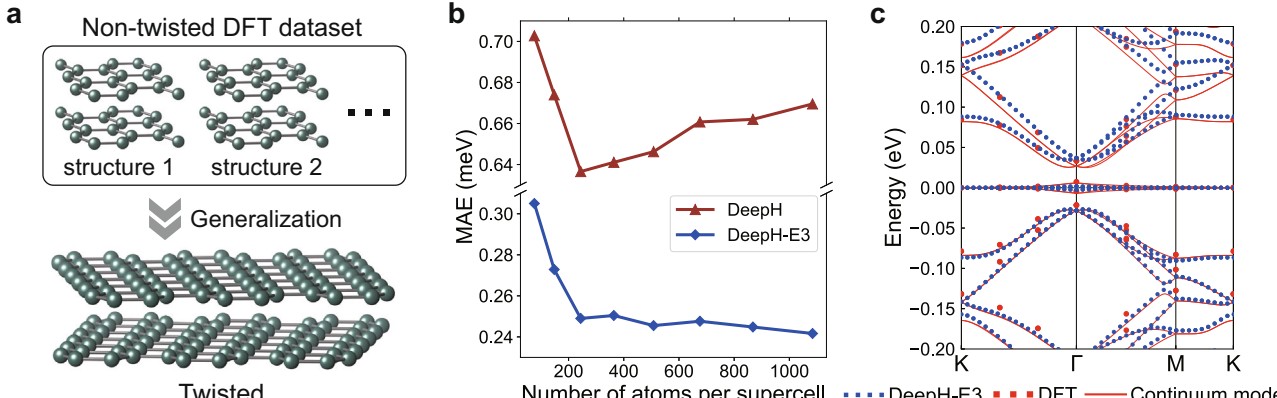

**Fig. 4 | Application of DeepH-E3 to study twisted bilayer graphene (TBG).**
**a** Workflow of DeepH-E3. The neural network model is first trained on DFT data of small, nontwisted, randomly perturbed structures and then generalized to study arbitrarily twisted structures without invoking DFT anymore. **b** Performance of DeepH-E3 vs. the original DeepH method[25] on studying TBGs of varying twist angle $\theta$. The averaged mean absolute errors (MAEs) of the DFT Hamiltonian are displayed for Moiré supercells of varied sizes. **c** Band structure of the magic-angle TBG

($\theta = 1.08°$, 11,164 atoms per supercell, structure relaxed by previous work[35]) computed by DeepH-E3, DFT, and continuum model[35]. Here the DFT benchmark calculations were performed with a different code using a plane-wave basis instead of an atomic-like basis and different pseudopotential, which could introduce numerical differences with respect to DeepH-E3. Source data are provided with this paper.

predicted by DeepH-E3 can accurately reproduce DFT results (Supplementary Fig. 1).

## Application to twisted vdW materials

Our deep learning method is particularly useful for studying the electronic structure of twisted vdW materials. This class of materials has attracted great interest for research and applications since their Moiré super periodicity offers a new degree of freedom to tune many-body interactions and brings in emergent quantum phenomena, such as correlated states[32], unconventional superconductivity[33], and higher-order band topology[34]. Traditionally, it is challenging to perform computationally demanding DFT calculations on large Moiré structures. However, this challenge could be largely overcome by DeepH-E3. One may train the neural network models by DFT data on small, nontwisted, randomly perturbed structures and predict the DFT Hamiltonian of arbitrarily twisted structures bypassing DFT via deep learning, as illustrated in Fig. 4a. This procedure demands much less computational resources than directly doing DFT calculations on large twisted superstructures.

Once the model is trained, it can be applied to study TBGs of varying twist angles. The performance is compared with that of DeepH. Test data includes DFT results for systems containing up to more than one thousand atoms per supercell. As summarized in Fig. 4b, DeepH-E3 significantly reduces the averaged MAEs of DFT Hamiltonian matrix elements by more than a factor of 2 as compared to DeepH, consistent with the above conclusion. Moreover, the MAEs reach ultralow values of 0.2−0.3 meV and gradually decrease with increasing Moiré supercell size (or decreasing twist angle). This demonstrates the good generalizability of DeepH-E3. The method is thus expected to be suitable for studying TBGs with small twist angles that are of current interest[35].

We take the magic-angle TBG with $\theta = 1.08°$ and 11,164 atoms per supercell as a special example. The discoveries of novel physics relevant to flat bands in this system have triggered enormous interest in investigating twisted vdW materials. Due to the large supercell, DFT study of magic-angle TBG is a formidable task, but DeepH-E3 can routinely study such kind of material systems in a particularly accurate and efficient way. As shown in Fig. 4c, the electronic bands of magic-angle TBG with relaxed structure computed by DeepH-E3 agree well with the published results obtained by DFT and low-energy effective continuum model[35]. The flat bands near the Fermi level are well reproduced. Some minor discrepancies appear away from the Fermi

level, which could be partially explained by the methodological difference: the benchmark work uses the plane-wave basis, whereas our work employs the atomic-like basis, and the pseudopotential used is also different. Detailed discussions about the influence of basis set and pseudopotential are included in Supplementary Note 2.

Most remarkably, DeepH-E3 has the capability to reduce the computational cost of studying these large material systems by several orders of magnitude. The DFT calculation (including structural relaxation) on magic-angle TBG performed in ref. 35 took around 1 month on about five thousand CPU cores. In contrast, the major computational cost of DeepH-E3 comes from neural network training. Typically, only a few hundreds of DFT training calculations are needed, and the training process usually takes tens of GPU hours, but all these are only required to be done once. After that, DFT Hamiltonian matrices can be constructed very efficiently via neural network inference. The process time is on the order of minutes by one GPU for magic-angle TBG, which grows linearly with Moiré supercell size. Generalized eigenvalue problems are solved for 60 bands near the Fermi level to obtain the band dispersion, which only requires about 8 min per **k**-point for magic-angle TBG using 64 CPU cores. The low computational cost and high accuracy of DeepH-E3 demonstrate its potential power in resolving the accuracy-efficiency dilemma of ab initio calculation methods, and it would be highly favorable to future scientific research.

## Study of twisted vdW materials with strong SOC

We have tested the performance of DeepH-E3 on studying twisted vdW materials with strong SOC, including twisted bilayers of bismuthene, $Bi_2Se_3$, and $Bi_2Te_3$. The latter two materials are more complicated, which include two quintuple layers and two kinds of elements (Fig. 5a for $Bi_2Te_3$). The strong SOC introduces additional complexity in their electronic structure problems. Despite all these difficulties, the capability of DeepH-E3 is not influenced to any extent. Our method reaches sub-meV accuracy in predicting DFT Hamiltonians of test material samples, including nontwisted and twisted structures of bismuthene, $Bi_2Se_3$, and $Bi_2Te_3$ bilayers. Impressively, the band structures predicted by DeepH-E3 match well with those obtained from DFT (Supplementary Fig. 3). Moreover, we observe the remarkable ability of our model to fit a tremendous amount of data with moderate model capacity and relatively small computational complexity. For instance, the neural network model is able to fit $2.8 \times 10^9$ nonzero complex-

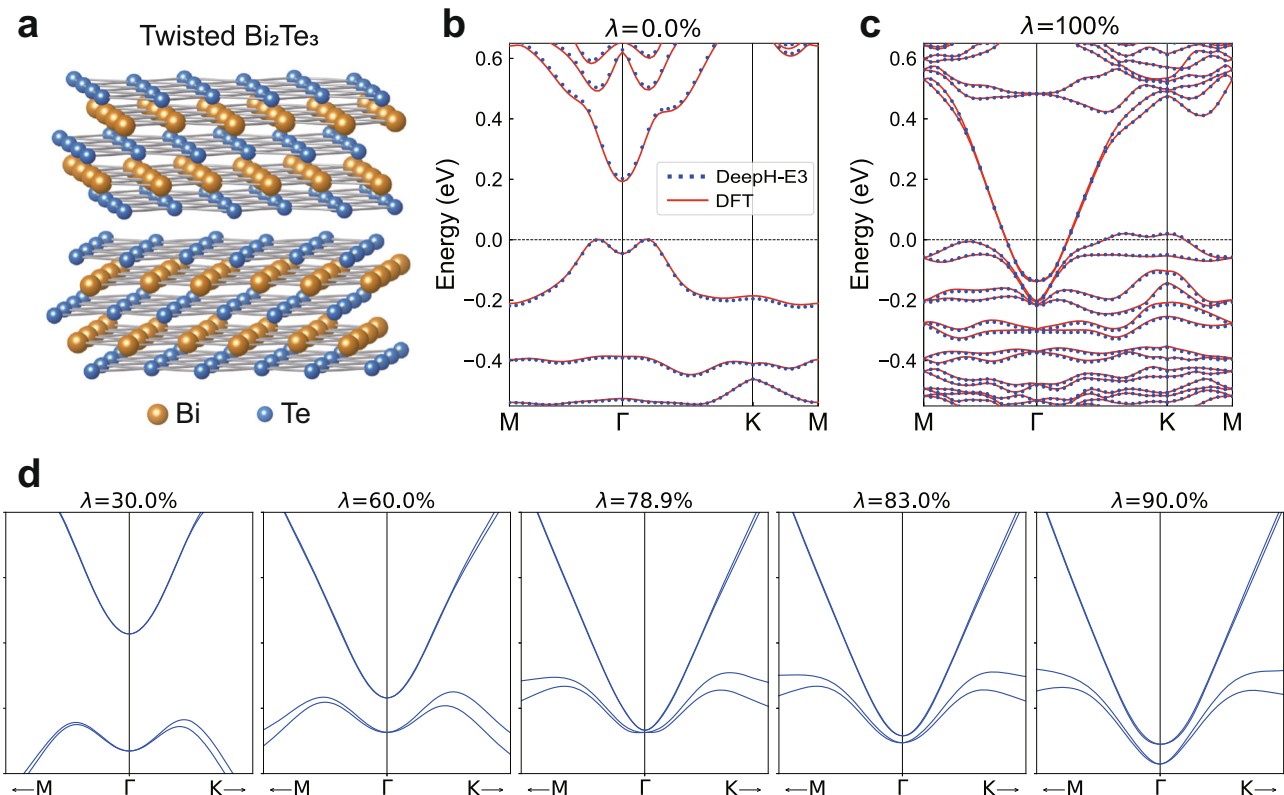

**Fig. 5 | Application of DeepH-E3 to study spin−orbit coupling (SOC) effects of twisted vdW materials. a** Schematic structure of twisted bilayer $Bi_2Te_3$. **b, c** Band structures of bilayer $Bi_2Te_3$ with twist angle $\theta = 21.8°$ predicted by DeepH-E3 without SOC ($\lambda = 0$) and with full SOC ($\lambda = 1$), compared to DFT results. **d** Evolution of band structure as a function of SOC strength predicted by DeepH-E3. Closing and reopening of the band gap at the $\Gamma$ point are visualized, indicating a topological quantum phase transition from $Z_2 = 0$ to $Z_2 = 1$ driven by SOC. Source data are provided with this paper.

valued Hamiltonian matrix elements in the dataset with about $10^5$ real parameters. The training time is about one day on a single GPU in order to reach sub-meV accuracy. More details are presented in Supplementary Note 3. Through these experiments, the capability of DeepH-E3 to represent the spin−orbital DFT Hamiltonian is well demonstrated.

In physics, the SOC can induce many exotic quantum phenomena, leading to emergent research fields of spintronics, unconventional superconductivity, topological states of matter, etc. Investigation of SOC effects is thus of fundamental importance to the research of condensed matter physics and materials science. The functionality of analyzing SOC effects is easily implemented by DeepH-E3. Specifically, we apply two neural network models to learn DFT Hamiltonians with full SOC ($\hat{H}_1$) and without SOC ($\hat{H}_0$) separately for the same material system. Then, we define a virtual Hamiltonian as a function of SOC strength ($\lambda$): $\hat{H}_\lambda = \hat{H}_0 + \lambda \hat{H}_{SOC}$, where $\hat{H}_{SOC} = \hat{H}_1 - \hat{H}_0$. By studying the virtual Hamiltonian at different $\lambda$, we can systematically analyze the influence of SOC effects on material properties.

As an example application, we employ the approach to investigate the topological properties of twisted bilayer $Bi_2Te_3$. DeepH-E3 can accurately predict the DFT Hamiltonian for both cases with or without SOC, as confirmed by band structure calculations using the predicted $\hat{H}_{DFT}$ (Fig. 5b, c). Herein the SOC is extremely strong as caused by the heavy elements in the material. Consequently, the band structure changes considerably when SOC is turned on. The evolution of band structure as a function of SOC strength (Fig. 5d) provides rich information on the SOC effects. Importantly, the band gap closes and reopens when increasing the SOC strength, indicating a topological quantum phase transition from $Z_2 = 0$ to $Z_2 = 1$. This is further confirmed by applying symmetry indicators based on Kohn-Sham orbital

analysis and by performing Brillouin-zone integration of Berry connection and curvature over all occupied states via the Fukui-Hatsugai-Suzuki formalism[36]. The topological invariant $Z_2$ turns out to be nonzero for the spin−orbital coupled system, suggesting that the twisted bilayer $Bi_2Te_3$ ($\theta = 21.8°$) is topologically nontrivial. As DeepH-E3 works well for varying twist angles, the dependence of band topology on twist angle can be systematically computed, which will enrich the research of twisted vdW materials.

## Discussion

Since the DFT Hamiltonian $\hat{H}_{DFT}$ transforms covariantly between reference frames, it is natural and advantageous to construct the mapping from crystal structure $\{\mathcal{R}\}$ to $\hat{H}_{DFT}$ in an explicitly equivariant manner. In this context, we have developed a general framework to represent $\hat{H}_{DFT}$ with a deep neural network DeepH-E3 that fully respects the principle of covariance even in the presence of SOC. We have presented the theoretical basis, code implementation, and practical applications of DeepH-E3. The method enables accurate and efficient electronic structure calculation of large-scale material systems beyond the scope of traditional ab initio approaches, opening possibilities to investigate rich physics and novel material properties at a particularly low computational cost.

However, as the structure becomes larger, it becomes increasingly difficult to diagonalize the Hamiltonian matrix in order to obtain wavefunction-related physical quantities. This difficulty, instead of the limitations of DeepH-E3 method itself, will eventually become the bottleneck of accurate electronic structure predictions. Nevertheless, benefiting from the sparseness of the DFT Hamiltonian matrix under localized atomic orbital basis, many efficient $O(N)$ algorithms with high parallel efficiency are available for studying large-scale systems (e.g.,

supercells including up to $10^7$ atoms[37]). A combination of the DeepH-E3 method with such efficient linear algebra algorithms will be a promising direction for future study.

The unique abilities of DeepH-E3, together with the general framework of incorporating symmetry requirements and physical insights into neural network model design, might find wide applications in various directions. For example, the method can be applied to build a material database for a diverse family of Moiré-twisted materials. For each kind of material, only one trained neural network model will be needed for all the twisted structures in order to have full access to their electronic properties, which is a great advantage for high throughput material discovery. Moreover, since the deep-learning method does not rely on periodic boundary conditions, 2D materials with incommensurate twist angles can also be investigated, making the ab initio study of quasi-crystal phases possible. In addition, we could go one step further by calculating the derivative of the electronic Hamiltonian with respect to atomic positions via automatic differentiation techniques. This enables deep-learning investigation of the physics of electron-phonon coupling in large-scale materials, which has the potential to outperform the computationally expensive traditional methods of frozen phonon or density functional perturbation theory[38]. Furthermore, one may combine the deep learning method with advanced methods beyond the DFT level, such as hybrid functionals, many-body perturbation theory, time-dependent DFT, etc. These important generalizations, if any of them are realized, would greatly enlarge the research scope of ab initio calculation.

## Methods

### Datasets

Data generated in this study is available in public repositories at Zenodo[39–41].

Monolayer graphene: The dataset is taken from ref. 25. The dataset consists of 450 graphene structures with $6 \times 6$ supercells, generated by ab initio molecular dynamics performed by the Vienna ab initio simulation package (VASP)[42], using the PBE[43] exchange-correlation functional and the projector-augmented wave (PAW) pseudopotentials[44,45]. The cutoff energy of the plane waves is 450 eV, and only the Γ point is used in our **k**-mesh. Five thousand frames are obtained at 300K with time step 1fs, and then one frame is taken out every 10 frames starting from the 500th frame. Thus, there are 450 structures in the dataset. The Hamiltonians for training are calculated with the OpenMX code using the PBE functional and norm-conserving pseudopotential with C6.0-$s2p2d1$ PAOs with $5 \times 5$ Γ-centered **k**-sampling. Here 6.0 denotes the orbital cutoff radius in Bohr, $s2p2d1$ means there are $2 \times 1 = 2$ $s$-orbitals, $2 \times 3 = 6$ $p$-orbitals, and $1 \times 5 = 5$ $d$-orbitals.

Monolayer MoS$_2$: The dataset is also taken from ref. 25. Five hundred structures with $5 \times 5$ supercells are generated by ab initio molecular dynamics performed by VASP with PAW pseudopotential and PBE functional. The cutoff energy of the plane waves is 450 eV, and only the Γ point is used in our **k**-mesh. One thousand frames are taken at 300K with time step 1fs. The first 500 unequilibrated structures are discarded, and the remaining 500 structures are taken into the dataset. The Hamiltonians for training are calculated with the OpenMX code using the PBE functional and norm-conserving pseudopotential with Mo7.0-$s3p2d2$ and S7.0-$s2p2d1$ PAOs with $5 \times 5$ Γ-centered **k**-sampling.

Bilayer graphene: The dataset is also taken from ref. 25. Three hundred structures with $4 \times 4$ nontwisted supercells are generated by a uniform shift of one of the two vdW layers and inserting random perturbations to atomic positions in the mean time. The perturbations are within 0.1 Å along three cartesian directions. The supercells are constructed from bilayer unit cell structures relaxed with VASP[42] using PBE functional with vdW interaction corrected by DFT-D3 method with Becke–Jonson damping[46]. The optimal interlayer spacing is found to be 3.35 Å. The Hamiltonians of the dataset and twisted structures are

all calculated with the OpenMX code using the PBE functional and norm-conserving pseudopotential with C6.0-$s2p2d1$ PAOs.

Bilayer bismuthene, Bi$_2$Se$_3$, and Bi$_2$Te$_3$: The same procedure is used to generate nontwisted $3 \times 3$ bilayer supercells. The numbers of structures are 576, 576, and 256 for bismuthene, Bi$_2$Se$_3$, and Bi$_2$Te$_3$, respectively, but only a randomly selected subset is used for training (details can be found in Supplementary Note 4). The interlayer spacing is 3.20 Å, 2.50 Å, and 2.61 Å for bismuthene, Bi$_2$Se$_3$, and Bi$_2$Te$_3$, respectively. The interlayer spacing is defined to be the vertical distance between the lowest atom in the upper layer and the highest atom in the lower layer. The Hamiltonians of the dataset and twisted structures are all calculated with the OpenMX code using the PBE functional and norm-conserving pseudopotential with Bi8.0-$s3p2d2$, Se7.0-$s3p2d1$ and Te7.0-$s3p2d2$ PAOs.

### Details of neural network models

All the neural network models presented in this article are trained by directly minimizing the mean-squared errors of the model output compared to the Hamiltonian matrices computed by DFT packages, and the reported MAEs are also obtained from comparing model output to the DFT results. All physical quantities of materials are derived from the output Hamiltonian matrix.

Some details of neural network building blocks are described here. The Gaussian basis is adapted from ref. 4, which is defined as:

$$\mathbf{e}_B(|\mathbf{r}_{ij}|)_n = \exp\left(-\frac{\left(|\mathbf{r}_{ij}| - r_n\right)^2}{2\Delta^2}\right), \quad (11)$$

where $r_n, n = 0, 1, \ldots$ are evenly spaced, with intervals equal to $\Delta$. The E3Linear layer is defined as:

$$\text{E3Linear}(\mathbf{x})_{cm}^{(l)} = \sum_{c'=1}^{n^{(l)}} W_{cc'}^{(l)} x_{c'm}^{(l)} + b_c^{(l)}, \quad (12)$$

where $W_{cc'}^{(l)}, b_c^{(l)}$ are learnable weights and biases, $b_c^{(l)} = 0$ for $l \neq 0$. In the gate layer, the $l = 0$ part of the input feature is separated into two parts, denoted as $x_{1c}^{(0)}$ and $x_{2c}^{(0)}$. Notice that the index $m$ is omitted because $l = 0$. The output feature is calculated by

$$\text{Gate}(\mathbf{x})_{cm}^{(l)} = \begin{cases} \phi_1(x_{1c}^{(0)}), & l = 0 \\ \phi_2(x_{2c}^{(0)})x_{cm}^{(l)}, & l \neq 0 \end{cases}. \quad (13)$$

Here $\phi_1$ and $\phi_2$ are activation functions. In this work, we use $\phi_1$=SiLU and $\phi_2$=Sigmoid following ref. 7.

The ENN is implemented with the e3nn library[31] in version 0.3.5 and PyTorch[47] in version 1.9.0. The Gaussian basis expansion used as input to the EquiConv layer has a length of 128. The fully connected neural network in the EquiConv layer is composed of two hidden layers, each with 64 hidden neurons, using the SiLU function as nonlinear activation and a linear layer as output. A description of neural network hyperparameters for each material system and their selection strategy can be found in Supplementary Note 4.

### Reporting summary

Further information on research design is available in the Nature Portfolio Reporting Summary linked to this article.

## Data availability

The datasets for monolayer graphene, monolayer MoS$_2$, bilayer graphene, and bilayer bismuthene are available in ref. 39. Dataset for bilayer Bi$_2$Se$_3$ is available in ref. 40. Dataset for bilayer Bi$_2$Te$_3$ is available in ref. 41. Instructions on reproducing the DeepH-E3 models on these datasets can also be found in the corresponding repositories. Source data are provided with this paper.

## Code availability

The code used in the current study is available at GitHub (https://github.com/Xiaoxun-Gong/DeepH-E3) and Zenodo[48].

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

## Acknowledgements

This work was supported by the Basic Science Center Project of NSFC (grant no. 52388201), the National Science Fund for Distinguished Young Scholars (grant no. 12025405), the National Natural Science Foundation of China (grant no. 11874035), the Ministry of Science and Technology of China (grant nos. 2018YFA0307100 and 2018YFA0305603), the Beijing Advanced Innovation Center for Future Chip (ICFC), and the Beijing Advanced Innovation Center for Materials Genome Engineering. R.X. was funded by the China Postdoctoral Science Foundation (grant no. 2021TQ0187).

## Author contributions

Y.X. and W.D. proposed the project and supervised X.G. and H.L. in carrying out the research, with the help of N.Z. and R.X. All authors discussed the results. Y.X. and X.G. prepared the manuscript with input from the other co-authors.

## Competing interests

The authors declare no competing interests.
