## [Peer Review File · Nature Communications]

REVIEWER COMMENTS

Reviewer #1 (Remarks to the Author):

This manuscript reports an extension of the same groups' previous DeepH approach to learning Hamiltonian matrix entries from DFT data which was published earlier this year. The new DeepH-E3 approach correctly accounts for the transformation laws satisfied by Hamiltonians under a change of basis, with a consequent increase in the accuracy of the model. Novelty also comes through the extension of the approach to capture spin orbit coupling. This makes the work interesting, however before a decision can be made on whether it warrants publication in a broad-interest journal like Nature Comms I have a number of specific concerns which must be addressed:

- The work does not make sufficient reference to appropriate recent developments in the field. For example, similar equivariant methods for Hamiltonians have been reported for molecular systems (DoI:10.1063/5.0072784) and in the condensed phase (DoI:10.1038/s41524-022-00843-2). The manuscript should be modified to place the work in the appropriate context.
- To ensure reproducibility of the results, parameters and code to evaluate the new approach must be made available to other scientists. The data access statement says this will be done following publication, but this is insufficient to allow me to give an informed opinion on the accuracy and quality of the work presented here.
- Insufficient details are given of the DFT calculations performed to generate the training data. As well as giving all the necessary parameters these data should be made available to enable reproducibility.

Reviewer #2 (Remarks to the Author):

The manuscript by Gong et al. introduces the non-trivial extension of a $E(3)$ -equivariant neural network representation of DFT Hamiltonians to the treatment of the spin-orbit coupling. The methodological approach is described clearly and thoroughly and the applications of the method are in general convincing.

I would strongly recommend this manuscript for publication in Nature Communications, but first I would like to see the following points addressed:

- The discrepancy between DFT (+continuum-mode) and DeepH-E3 for the twisted bilayer graphene are attributed to the different basis sets and pseudopotentials. Since these errors are quantitatively small but qualitatively relevant, it would be good to investigate the dependence of the NN results as function of the changing the basis set and pseudos.
- the data and code are promised to become available upon publication. This is very vague. Following the FAIR-data paradigm, The authors should specify on which platform (e.g., zenodo, github, NOMAD) these items will be made available. In particular, the DFT data should be clearly indexed as training/test in order to facilitate the reproducibility of the results.

Reviewer #3 (Remarks to the Author):

Summary.

In this paper, the authors propose an $E(3)$ -equivariant neural network framework (DeepH-E3) to learn the density functional theory (DFT) Hamiltonian as a function of material structure in the presence of spin-orbit coupling (SOC). They show that ENNs can be trained on small-sized materials systems to predict the electronic structure of larger materials systems. Furthermore, they extend existing ENN frameworks to incorporate electronic spin and SOC. Existing ENN architectures only allowed for integer spin values $\ell = 0, 1, 2, \dots$, yet the authors use transformation rules in the final output layer to include electron spin $\ell = 1/2$. Additionally, the spin-orbital Hamiltonian matrix can have complex values, so the authors show how to preserve

equivariance with predicting real and imaginary parts for integer ℓ and $\ell = 1/2$. This represents a noteworthy contribution to the use of ENNs for learning DFT Hamiltonians.

The authors first validate performance on a dataset of DFT supercell calculation results of monolayer graphene and MoS₂. They measure performance by the mean absolute errors (MAEs) of Hamiltonian matrix elements averaged over atom pairs. MAEs are reduced by a factor of 2 or more when compared to DeepH (a recent framework using rotation-invariant neural networks). Next, they show that one can use DeepH-E3 to study the electronic structure of twisted vdW materials, a problem that is challenging due to the computational cost of performing DFT calculations on large Moiré structures. The trained model is applied to study twisted bilayer graphene (TBGs) of varying twist angles, and the electronic bands agree well with the published results obtained by SFT and the low-energy effective continuum model. DeepH-E3 also reduces the computational cost to study these large material systems by several orders of magnitude (eg one month to do a DFT calculation vs neural-network inference post training for DeepH-E3).

They also test the performance of DeepH-E3 on studying twisted vdW materials with strong SOC, and find that it reaches sub-meV accuracy in predicting the DFT Hamiltonian of test material samples. Finally they employ the DeepH-E3 models to study the effects of SOC through implementing two neural network models to learn DFT Hamiltonians with full SOC and without SOC separately for the same material system. They then study the virtual Hamiltonian composed of the difference between the two as a function of SOC strength (λ). They employ this approach to investigate the topological properties of twisted bilayer Bi₂Te₃. The authors conclude that DeepH-E3 represents a promising approach for deep-learning investigation of the physics of large-scale materials as well as accelerating ab initio calculations.

Comments.

This is a well-written and well-organized paper that does represent original contributions to the use of equivariant neural networks for learning DFT Hamiltonians. The work does support the conclusions and claims. The methodology is sound with enough detail provided for the work to be reproduced. I would recommend publication of this paper. Below are a few comments.

Introduction.

In paragraph 1, the authors introduce the DeepH model as a deep neural network representation of the DFT Hamiltonian. It may make more sense to first generally introduce deep learning for physical systems, then discuss ENN models, and then emphasize that you are integrating ENN models with DeepH (ie don't introduce DeepH until the second or third paragraph).

Results.

Realization of Equivariance.

I would recommend restructuring this section to align better with Figure 2. First discuss the transformation rules without SOC and how the input, output, and internal features of ENNs transform according to Eq. (3). Next, discuss the issues raised in the construction of the spin-orbital Hamiltonian and the resolution of these issues. Eq. (3) is missing a prime on x_{lm} .

Neural Network Architecture of DeepH-E3.

It would be good to justify why E3LayerNorm is needed.

Capability of DeepH-E3.

Throughout the results section, the authors benchmark through comparing to DeepH. Are there any other models that would be beneficial to compare to/why is DeepH a reliable benchmark to use? For example, see "Equivariant analytical mapping of first principles Hamiltonians to accurate and transferable materials models" (Zhang, L., Onat, B., Dusson, G. et al., 2022, <https://doi.org/10.1038/s41524-022-00843-2>), which offers an equivariant scheme to construct predictive models for Hamiltonian matrices in the atomic orbital representation without using deep learning. Also see "SE(3)-equivariant prediction of molecular wavefunctions and electronic densities" (Unke, O. T., Bogojeski, M., Gastegger, M., et al., NeurIPS,

<https://doi.org/10.48550/arXiv.2106.02347>) applying ENNs to predicting Hamiltonians of molecular systems. The authors should justify the choice of only comparing with DeepH/perhaps present more work in this area.

Although the datasets are outlined in the Methods section, it may be useful to introduce them at the beginning of this section to specify how the Hamiltonians are obtained (eg they were obtained using VASP). Here it would be relevant to introduce potential issues with using a plane-wave centered code to compare with your method using the localized atomic orbital basis. It would be useful to justify why you use VASP rather than a different DFT code.

In terms of structure, this section is a bit long/disorganized. You could split the section up based on applications. It would be good to have a separate paragraph at the end discussing emphasizing the improvements in computational cost using DeepH-E3 as this is currently repeated throughout this section.

Discussion.

No comments.

Methods.

Datasets.

Details of neural network models.

It would be good to include details of hyperparameter tuning for the ENNs used. I see some details in the supplementary note but it is unclear what the optimal hyperparameters are/how these were chosen (eg did you use default parameters present in ENN? How did you search for these parameters? (grid search, random search, etc)

RESPONSE TO REVIEWERS' COMMENTS

Reviewer #1:

This manuscript reports an extension of the same group's previous DeepH approach to learning Hamiltonian matrix entries from DFT data which was published earlier this year. The new DeepH-E3 approach correctly accounts for the transformation laws satisfied by Hamiltonians under a change of basis, with a consequent increase in the accuracy of the model. Novelty also comes through the extension of the approach to capture spin orbit coupling. This makes the work interesting, however before a decision can be made on whether it warrants publication in a broad-interest journal like Nature Comms I have a number of specific concerns which must be addressed:

Response: We gratefully thank the referee for the careful review of our manuscript. We also thank the referee for recognizing the accomplishments of our work in incorporating symmetry requirements in neural network models and the novelty in extension to spin-orbit coupled systems. In the following we will make a point-to-point response to the comments raised.

Comment 1: The work does not make sufficient reference to appropriate recent developments in the field. For example, similar equivariant methods for Hamiltonians have been reported for molecular systems (DoI:10.1063/5.0072784) and in the condensed phase (DoI:10.1038/s41524-022-00843-2). The manuscript should be modified to place the work in the appropriate context.

Response: We thank the referee for suggesting references. The mentioned two references are about equivariant methods of machine learning Hamiltonians, which are truly relevant to the research topic. In the Introduction part, we added a sentence "*People have already made attempts to model the Hamiltonian using equivariant methods*", cited the two references as Ref. [27] [J. Chem. Phys. 156, 014115 (2022)] and Ref. [28] [npj Comput. Mater. 8, 158 (2022)], and discussed them explicitly: "*Nigam et al. used rotationally equivariant N-center features in the kernel ridge regression method to fit molecular Hamiltonians [27]. Zhang et al. proposed an equivariant scheme to parameterize the Hamiltonians of crystals based on the atomic cluster expansion descriptor [28].*"

Comment 2: To ensure reproducibility of the results, parameters and code to evaluate the new approach must be made available to other scientists. The data access statement says this will be done following publication, but this is insufficient to allow me to give an informed opinion on the accuracy and quality of the work presented here.

Response: We fully agree with the referee that all data and code in scientific works should be transparent and accessible to fellow scientists. In order to comply with the editorial policies of Nature Communications, we will submit our code alongside the revised manuscript and they will be accessible to all reviewers during the review process, but it will only be available to public when the manuscript is published. The dataset will be uploaded to a zenodo public repository, which will include all the structure information and the corresponding Hamiltonian matrices for training. We will also upload detailed description to reproduce the neural network models based on the datasets using our code.

Comment 3: Insufficient details are given of the DFT calculations performed to generate the training data. As well as giving all the necessary parameters these data should be made available to enable reproducibility.

Response: We thank the referee for the helpful comment. To make sure that our results can be reproduced by others, we revised the “Dataset” subsection in Methods, and presented all the necessary details of DFT calculations explicitly for each material systems we considered. In addition, all datasets used in this work will be made available upon publication, so any group interested in the data will easily access them from the website.

Reviewer #2:

The manuscript by Gong et al. introduces the non-trivial extension of a E(3)-equivariant neural network representation of DFT Hamiltonians to the treatment of the spin-orbit coupling.

The methodological approach is described clearly and thoroughly and the applications of the method are in general convincing.

I would strongly recommend this manuscript for publication in Nature Communications, but first I would like to see the following points addressed:

Response: We gratefully thank the referee for his/her careful review of our manuscript. We also sincerely appreciate the referee for the high recognition and constructive comments on our work. Below we give a point-to-point response to the comments raised.

Comment 1: The discrepancy between DFT (+continuum-model) and DeepH-E3 for the twisted bilayer graphene are attributed to the different basis sets and pseudopotentials. Since these errors are quantitatively small but qualitatively relevant, it would be good to investigate the dependence of the NN results as function of the changing the basis set and pseudos.

Response: We thank the referee for the helpful suggestion. To check the influence of basis set and pseudo potential on the neural network results, we calculated the electronic structure of a twist bilayer graphene (twist angle $\theta = 6.01^\circ$) by VASP and OpenMX as shown in Fig. R1. We prefer using this indirect way of checking instead of directly performing neural network studies, due to the following reasons: (i) The target quantity of deep learning is the DFT Hamiltonian under localized basis, which could be directly generated by the localized-basis code (like OpenMX) but not by the plane-wave code (like VASP). (ii) The OpenMX code employed in this work only supports a fixed set of norm-conserving pseudopotential distributed alongside the code package. (iii) DeepH-E3 can accurately reproduce DFT Hamiltonians of OpenMX, as systematically demonstrated in this work. Therefore, we believe that the comparison of electronic structures calculated by VASP and OpenMX will fit the purpose.

As shown in Fig. R1, our test calculations indicate that the use of different basis sets and pseudo potentials (VASP vs. OpenMX) has minor influence on the calculated electronic bands, especially those near the Fermi level, at least for this particular material (i.e., twisted bilayer graphene with $\theta = 6.01^\circ$). As the size of material system increases, the calculation gets more and more difficult to converge and the corresponding numerical error typically grows, which might enhance the band-structure discrepancies between the two approaches. This, however, is difficult to check for the magic-angle twisted bilayer graphene, because we cannot do the benchmark calculation directly by using the OpenMX code as limited by the huge computational cost.

Figure R1. Band structures of twisted bilayer graphene with twist angle $\theta = 6.01^\circ$ calculated by VASP and OpenMX.

As a response, we added Supplementary Note 2 in the supplementary material, presented the above discussion in it, and included Fig. R1 as supplemental Fig. S2.

Comment 2: The data and code are promised to become available upon publication. This is very vague. Following the FAIR-data paradigm, the authors should specify on which platform (e.g., zenodo, github, NOMAD) these items will be made available. In particular, the DFT data should be clearly indexed as training/test in order to facilitate the reproducibility of the results.

Response: We thank the referee for the helpful suggestion. In order to comply with the editorial policies of Nature Communications, we will submit our code alongside the revised manuscript and they will be accessible to all reviewers during the review process, but it will only be available to public when the manuscript is published. The dataset will be uploaded to a zenodo public repository, which will include all the structure information and the corresponding Hamiltonian matrices for training. We will also upload detailed description to reproduce the neural network models based on the datasets using our code.

Reviewer #3:

Summary.

In this paper, the authors propose an E(3)-equivariant neural network framework (DeepH-E3) to learn the density functional theory (DFT) Hamiltonian as a function of material structure in the presence of spin-orbit coupling (SOC). They show that ENNs can be trained on small-sized materials systems to predict the electronic structure of larger materials systems. Furthermore, they extend existing ENN frameworks to incorporate electronic spin and SOC. Existing ENN architectures only allowed for integer spin values $l = 0, 1, 2, \dots$, yet the authors use transformation rules in the final output layer to include electron spin $l = 1/2$. Additionally, the spin-orbital Hamiltonian matrix can have complex values, so the authors show how to preserve equivariance with predicting real and imaginary parts for integer l and $l = 1/2$. This represents a noteworthy contribution to the use of ENNs for learning DFT Hamiltonians.

The authors first validate performance on a dataset of DFT supercell calculation results of monolayer graphene and MoS₂. They measure performance by the mean absolute errors (MAEs) of Hamiltonian matrix elements averaged over atom pairs. MAEs are reduced by a factor of 2 or more when compared to DeepH (a recent framework using rotation-invariant neural networks). Next, they show that one can use DeepH-E3 to study the electronic structure of twisted vdW materials, a problem that is challenging due to the computational cost of performing DFT calculations on large Moire structures. The trained model is applied to study twisted bilayer graphene (TBGs) of varying twist angles, and the electronic bands agree well with the published results obtained by SFT and the low-energy effective continuum model. DeepH-E3 also reduces the computational cost to study these large material systems by several orders of magnitude (eg one month to do a DFT calculation vs neural-network inference post training for DeepH-E3).

They also test the performance of DeepH-E3 on studying twisted vdW materials with strong SOC, and find that it reaches sub-meV accuracy in predicting the DFT Hamiltonian of test material samples. Finally they employ the DeepH-E3 models to study the effects of SOC through implementing two neural network models to learn DFT Hamiltonians with full SOC and without SOC separately for the same material system. They then study the virtual Hamiltonian composed of the difference between the two as a function of SOC strength (λ). They employ this approach to investigate the topological properties of twisted bilayer Bi₂Te₃. The authors conclude that DeepH-E3 represents a promising approach for deep-learning investigation of the physics of large-scale materials as well as accelerating ab initio calculations.

Comments.

This is a well-written and well-organized paper that does represent original contributions to the use of equivariant neural networks for learning DFT Hamiltonians. The work does support the conclusions and claims. The methodology is sound with enough detail provided for the work to be reproduced. I would recommend publication of this paper. Below are a few comments.

Response: We gratefully thank the referee for the careful review of our manuscript. We also sincerely appreciate the referee for the well-written summary, insightful suggestions, and high recognition on our work. Below we will give a point-to-point response to the comments raised.

Comment 1: In paragraph 1, the authors introduce the DeepH model as a deep neural network representation of the DFT Hamiltonian. It may make more sense to first generally introduce deep learning for physical systems, then discuss ENN models, and then emphasize that you are integrating ENN models with DeepH (ie don't introduce DeepH until the second or third paragraph).

Response: We thank the referee for the helpful suggestion. Following the referee's suggestion, we first generally introduced deep learning for physical systems, then discussed ENN models, introduced DeepH afterwards, and finally emphasized that we will try to integrate ENN models with DeepH in the Introduction part. As you can find by the colored revisions, we have significantly restructured the Introduction part to make it logically more straightforward.

Comment 2: I would recommend restructuring this section to align better with Figure 2. First discuss the transformation rules without SOC and how the input, output, and internal features of ENNs transform according to Eq. (3). Next, discuss the issues raised in the construction of the spin-orbital Hamiltonian and the resolution of these issues. Eq. (3) is missing a prime on χ_{lm} .

Response: Following the referee's suggestion, we moved the part discussing issues related to SOC to a new section "Equivariance of the spin-orbital Hamiltonian" after the original section "Realization of equivariance". The discussion will logically align better with Fig. 2 and be easier to understand. We thank the referee for the valuable suggestion. We also thank the referee for pointing out the typo. It is corrected in the revised manuscript.

Comment 3: It would be good to justify why E3LayerNorm is needed.

Response: We thank the referee for raising this point. Batch normalization and layer normalization are widely adapted techniques in machine learning to make the training process of neural networks efficient and stable. For DeepH-E3, batch normalization is not very helpful because every single crystalline material structure for training has a large amount of Hamiltonian matrix elements and thus the batch size is usually chosen to be small. In this situation, it will be useful to develop a normalization scheme using the layer statistics while fully respecting the equivariance of the feature vectors. In our experiments, we find that the introduction of E3LayerNorm stabilizes the training process (Fig. R2), thus enables higher learning rates, which is not only beneficial for the optimization of the neural network parameters but also improves the generalization ability of the model.

Figure R2. Comparison of the training process with or without E3LayerNorm, using the monolayer MoS₂ dataset. The vertical axis measures the MSE of Hamiltonian matrix elements in unit of eV². Computational details are described in Supplementary Note 5.

As a response, we added Supplementary Note 5 in the supplementary material, presented the above discussion in it, and included Fig. R2 as supplemental Fig. S4.

Comment 4: Throughout the results section, the authors benchmark through comparing to DeepH. Are there any other models that would be beneficial to compare to/why is DeepH a reliable benchmark to use? For example, see “Equivariant analytical mapping of first principles Hamiltonians to accurate and transferable materials models” (Zhang, L., Onat, B., Dusson, G. et al., 2022, <https://doi.org/10.1038/s41524-022-00843-2>), which offers an equivariant scheme to construct predictive models for Hamiltonian matrices in the atomic orbital representation without using deep learning. Also see “SE(3)-equivariant prediction of molecular wavefunctions and electronic densities” (Unke, O. T., Bogojeski, M., Gastegger, M., et al., NeurIPS, <https://doi.org/10.48550/arXiv.2106.02347>) applying ENNs to predicting Hamiltonians of molecular systems. The authors should justify the choice of only comparing with DeepH/perhaps present more work in this area.

Response: We thank the referee for the expert comment. We chose to compare DeepH-E3 with DeepH based on the following considerations: (i) The only available code that can use deep neural networks to learn DFT Hamiltonians of crystalline materials and make predictions on large-size materials by learning from smaller ones, as far as we know, is DeepH. (ii) DeepH-E3 and DeepH share essentially the same neural network architecture, except that they apply distinct strategies to deal with the rotational covariance of Hamiltonian. The advantages of using equivariant neural networks thus can be clearly demonstrated by comparing the performance of these two methods using the same training datasets.

Zhang et al. proposed an equivariant scheme to parameterize the Hamiltonians of crystals based on

the atomic cluster expansion descriptor [npj Comput. Mater. 8, 158 (2022)]. The applied equivariant scheme is interesting from the theoretical point of view. The method also has high computational efficiency. However, its prediction accuracy is not comparable with deep-learning methods, whose predicted band structure show significant discrepancies with DFT results for the relatively simple materials (see Fig. 5 of the reference). The reference only used 4 structures for training, which, of course, cannot be compared to a deep neural-network based approach on a fair basis. We thus prefer not comparing with this work. Instead, we cited and briefly discussed the reference in the Introduction part.

The work of Unke et al. [arXiv:2106.02347] was originally cited in the manuscript, which developed an ENN method named PhiSNet for predicting the Hamiltonian of molecular systems. This work achieved significant improvements over the predecessor SchNOrb [Nat. Commun. 10, 5024 (2019)], which employs data augmentation to inexplicitly introduce rotational equivariance. The performance of DeepH has been compared with these two methods before [Nat. Comput. Sci. 2, 367 (2022)]. Here we chose a molecular data set of ethanol (C_2H_6O), and performed deep learning studies by DeepH-E3.

The comparison of performance with DeepH, SchNOrb, and PhiSNet are presented in Tab. R1. The obtained MAE of DeepH-E3 is larger than DeepH and PhiSNet. It should be noted that the neural network models of DeepH-E3 are not fully optimized in our preliminary test. The performance of DeepH-E3 is expected to be better than DeepH, but the test is out of expectation. Moreover, DeepH-E3 is designed for studying large-size material systems. For instance, the neural network ensures strict locality by employing the message passing scheme in order to reach linear scaling and good generalizability to large-scale materials. Such kind of consideration, however, is not necessary for neural networks for small molecules. This might explain why DeepH-E3 is more efficient and less accurate than PhiSNet in the study of molecules. Therefore, careful design and optimization would be required if DeepH-E3 is to be applied to molecular systems.

Table R1. Comparison of the performance of DeepH-E3 with other deep-learning methods on the ethanol (C_2H_6O) dataset. Performance data of other methods are extracted from Ref. [Nat. Comput. Sci. 2, 367 (2022)].

	Inference time (ms)	Number of parameters	Test MAE (meV)
DeepH-E3	3.5	$\sim 10^6$	1.100
DeepH	2.3	$\sim 10^6$	0.601
SchNOrb	–	$\sim 10^7$	5.099
PhiSNet	27	$\sim 10^7$	0.331

Comment 5: Although the datasets are outlined in the Methods section, it may be useful to introduce them at the beginning of this section to specify how the Hamiltonians are obtained (eg they were obtained using VASP). Here it would be relevant to introduce potential issues with using a plane-wave centered code to compare with your method using the localized atomic orbital basis. It would be useful to justify why you use VASP rather than a different DFT code.

Response: To take advantage of the nearsightedness property of the Hamiltonian under localized basis, all the DFT Hamiltonian matrices used for deep learning in this work are computed by the OpenMX code using the pseudo-atomic orbital (PAO) basis. We added a sentence in the beginning paragraph of the section “Capability of DeepH-E3” to explicitly specify the fact.

Because the plane-wave methods are typically more accurate than the localized basis ones for the calculation of total energy, we applied the most widely used plane-wave code VASP to perform structure relaxation. However, the DFT Hamiltonian under plane-wave basis does not have the nearsightedness property. We thus did not use VASP to compute the DFT Hamiltonian for deep learning.

Comment 6: In terms of structure, this section is a bit long/disorganized. You could split the section up based on applications. It would be good to have a separate paragraph at the end discussing emphasizing the improvements in computational cost using DeepH-E3 as this is currently repeated throughout this section.

Response: We thank the referee for the constructive suggestion. To make the section “Capability of DeepH-E3” easier to understand, we split it into several shorter sections and described the experiments on one kind of material system in each section. Moreover, the referee suggested to put the discussion on computational cost of DeepH-E3 into a separate paragraph. Considering that the advantage of using DeepH-E3 is well illustrated by the study of twisted vdW materials, we chose to put discussions about the improvement of computational cost by DeepH-E3 into the last paragraph of the section “Application to twisted vdW materials”.

Comment 7: It would be good to include details of hyperparameter tuning for the ENNs used. I see some details in the supplementary note but it is unclear what the optimal hyperparameters are/how these were chosen (eg did you use default parameters present in ENN? How did you search for these parameters? (grid search, random search, etc)

Response: According to the referee’s suggestion, we have summarized the hyperparameters in supplemental Tab. S2. Any interested reader could easily look up the hyperparameters of any model for the different materials that are studied. In our experiments, we find that most of the hyperparameters have minor influence on the final model performance as long as they are within a reasonable range. We select the optimal hyperparameters by the following strategies. On the learning rate, we find that too large learning rate might cause instability in the training process: the loss function might suddenly blow up to a very high value. Too small learning rate will make the training slow and increase the probability of overfitting. Therefore, we first restrict the learning rate within a reasonable range to avoid these two situations, and then select the learning rate that gives the best model. On the batch size, it is usually limited by the computer memory. One single crystalline structure can have as many as $\sim 10^6$ Hamiltonian matrix elements, so the batch size is usually chosen to be 1. When studying relatively simple materials, the batch sizes will be increased in order to

speed up the training process. On hyperparameters that are related to the expressive power of the model, such as the length of the internal vertex and edge features, the maximum angular momentum of spherical harmonics and the number of message-passing layers, there is a tradeoff between the network performance and the training time. Usually, more complex neural networks could realize smaller prediction error. We choose those parameters to make sure that we could produce the best results within an affordable training time.

As a response, we added Supplementary Note 4 in the supplementary material, presented the above discussion in it, and added supplemental Tab. S2 to summarize network hyperparameters.

Summary of changes (Revisions are colored blue in the revised manuscript.)

- The first two paragraphs of the Introduction part were restructured following the suggestion of Reviewer #3.
- In the third paragraph on page 1, three sentences together with two references (i.e., Refs. [27,28]) were added.
- The part discussing issues related to SOC in the section “Realization of equivariance” on page 3 were moved to a new section “Equivariance of the spin-orbital Hamiltonian” on page 4.
- In the section “Capability of DeepH-E3” on page 6, one sentence “*All the DFT Hamiltonian matrices used for deep learning in this work are computed by the OpenMX code using the PAO basis*” was added.
- The original section “Capability of DeepH-E3” was split up into four sections: “Capability of DeepH-E3”, “Study of monolayer graphene and MoS₂”, “Application to twisted vdW materials”, and “Study of twisted vdW materials with strong SOC”.
- Discussions about improvement of computational cost by DeepH-E3 was put into the last paragraph of the section “Application to twisted vdW materials” on pages 7-8.
- The “Datasets” section in Methods on page 10 was revised to present more details of DFT calculations.
- Specific statements on code availability and data availability were made.
- Three Supplementary Notes (Notes 2, 4, 5), two supplemental figures (Figs. S2 and S4) and one supplemental table (Tab. S2) were added in the supplementary material. One sentence was added on page 8 to cite Supplementary Note 2. One sentence was added on page 11 to cite Supplementary Note 4. Description of neural network hyperparameters on page 11 was moved to supplemental Tab. S2. One sentence was added on page 5 to cite Supplementary Note 5.

REVIEWERS' COMMENTS

Reviewer #1 (Remarks to the Author):

I have reviewed the revised manuscript in detail. The authors have responded fully to all the comments raised by myself and the other reviewers. I am now happy to recommend publication.

Reviewer #2 (Remarks to the Author):

The authors have satisfactorily addressed my previous comments. In particular, the datasets uploaded on zenodo (see data availability section) are thoroughly structured so that it is possible to reproduce the results. I therefore recommend the present version of the manuscript for publication on Nature Communications.

Reviewer #3 (Remarks to the Author):

My comments and questions have been addressed and I recommend the work for publication.